# Polyamine Immunometabolism: Central Regulators of Inflammation, Cancer and Autoimmunity

**DOI:** 10.3390/cells11050896

**Published:** 2022-03-05

**Authors:** Tzu-yi Chia, Andrew Zolp, Jason Miska

**Affiliations:** 1Department of Neurological Surgery, Feinberg School of Medicine, Northwestern University, 676 N St. Clair, Suite, Chicago, IL 60611, USA; tzu-yichia2023@u.northwestern.edu (T.-y.C.); andrew.zolp@law.northwestern.edu (A.Z.); 2Northwestern Medicine Malnati Brain Tumor Institute of the Lurie Comprehensive Cancer Center, Feinberg School of Medicine, Northwestern University, Chicago, IL 60611, USA

**Keywords:** polyamines, autoimmunity, cancer, myeloid cells

## Abstract

Polyamines are ubiquitous, amine-rich molecules with diverse processes in biology. Recent work has highlighted that polyamines exert profound roles on the mammalian immune system, particularly inflammation and cancer. The mechanisms by which they control immunity are still being described. In the context of inflammation and autoimmunity, polyamine levels inversely correlate to autoimmune phenotypes, with lower polyamine levels associated with higher inflammatory responses. Conversely, in the context of cancer, polyamines and polyamine biosynthetic genes positively correlate with the severity of malignancy. Blockade of polyamine metabolism in cancer results in reduced tumor growth, and the effects appear to be mediated by an increase in T-cell infiltration and a pro-inflammatory phenotype of macrophages. These studies suggest that polyamine depletion leads to inflammation and that polyamine enrichment potentiates myeloid cell immune suppression. Indeed, combinatorial treatment with polyamine blockade and immunotherapy has shown efficacy in pre-clinical models of cancer. Considering the efficacy of immunotherapies is linked to autoimmune sequelae in humans, termed immune-adverse related events (iAREs), this suggests that polyamine levels may govern the inflammatory response to immunotherapies. This review proposes that polyamine metabolism acts to balance autoimmune inflammation and anti-tumor immunity and that polyamine levels can be used to monitor immune responses and responsiveness to immunotherapy.

## 1. Introduction

Polyamines are highly charged aliphatic polycations present in all living organisms. Polyamines are so fundamental to mammalian biology that depletion leads to total arrest in translation and proliferation of all cells [1]. One explanation for polyamines’ ubiquitous role is their positive charge, which gives them a proclivity to bind to DNA, RNA, and nuclear proteins. The mechanisms by which polyamines promote cellular proliferation are manifold. Polyamines facilitate proliferative processes by modifying the chromosomal structure, influencing protein/DNA interactions, and likely through other mechanisms that are still unknown [2]. Another role in proliferation for polyamines is as a substrate in the hypusination of the transcription factor EIF5a. EIF5a is an ancient transcription factor found across all archaea and eukaryotes, and the only polyamine-covalent modification described to date [3], key in regulating cellular translation [4,5]. They also play an important role in the apoptotic processes within a cell. Early studies identified polyamines promote mitochondrial cytochrome c release [6], and more recent work has demonstrated that p53-mediated ferroptosis is directly controlled by polyamine catabolism [7]. Therefore, polyamines are central to the proliferation and survival of all cells in most lifeforms.

Beyond their fundamental roles in cellular proliferation, transcriptional activation and cell survival, polyamines also support roles in diverse biological processes in the CNS. Due to the high polyamine concentration seen in the mammalian brain, research has focused on their role in the CNS [8]. Evidentiary for their influence on global CNS function, increased levels of polyamines are associated with suicidal completion [9]. In fact, recent studies have indicated that polyamines can promote longevity and cognitive function [10,11,12]. In these studies, the authors demonstrate spermidine administration from exogenous sources, including the diet [12], leads to enhanced neurocognitive function through the induction of protective autophagy and increased mitochondrial function. In maintaining CNS homeostasis, they mainly function as regulators on neurotransmitter metabolism [13] or—in the case of agmatine—may be a neurotransmitter in and of itself [14]. Furthermore, the positive charge of polyamines regulates ion channels by inhibiting inward rectifying potassium channels [15] that can modulate N-methyl- aspartate receptor function [16].

In mammalian cells, there have been three major polyamines identified: putrescine, spermidine, and spermine [17]. The metabolic precursor of all polyamines is ornithine, which is exclusively generated by the catabolism of arginine by the enzymes arginase (Arg1 and Arg2) and Glycine Amidinotransferase (Gatm) [18]. Ornithine is decarboxylated by the enzyme Ornithine decarboxylase 1 (Odc1), a primordial enzyme that catalyzes the production of putrescine through the conversion of ornithine. Odc1 is directly regulated through the actions of activating and inhibiting enzymes Azin and Oaz, respectively [19]. The longer chain polyamines, spermidine and spermine, are produced through the transfers of an aminopropyl moiety of decarboxylated S-adenosylmethionine (dcSAM) to the secondary amines of putrescine [20].

Polyamine metabolism is tightly controlled within cells and can be catabolized by two independent metabolic processes. In one pathway, spermine can be converted directly to spermidine through the enzyme spermine oxidase [21], which generates reactive oxygen species (ROS) as a byproduct of polyamine catabolism. A second pathway is regulated by the enzyme spermidine/spermine-N-acetyltransferase (SAT1), producing N-acetylspermidine and N-acetylspermine, leading to its expulsion from the cell [22]. The acetylated polyamines can then form putrescine through oxidative deamination reactions catalyzed by polyamine oxidase, which also generate ROS as a byproduct. Importantly, polyamines can be acquired from the extracellular environment by all domains of life and rescue the proliferation of cells where synthesis is inhibited [23].

Beyond the myriad of roles that polyamines have in basic cellular biology, polyamines have an ever-increasing appreciation for their role in immune biology. The basic premise of polyamines influencing immunity can be traced back to a seminal study in Nature from 1977 [24]. In this pioneering work, they found that exogenous polyamine administration inhibited both innate and adaptive immune responses of murine splenocytes. At the time, they did not quite know what polyamines were inhibiting or why, but they knew the levels were increased in environments where the immune system was suppressed (fetal tissue, malignant tissues, and reproductive fluids). After 45 years of intensive research and thousands of studies published, we are still learning new and surprising ways these small positively charged molecules exert influence on both autoimmunity and cancer. While the broad strokes of this initial observation remain true (i.e., polyamines are immunosuppressive), there are still significant discrepancies between cell types and disease states that have furthered our understanding of polyamine function in immunity. Below, we will discuss the current landscape of polyamines in autoimmunity, immune suppression, and finally how immunotherapies may regulate these processes.

## 2. The Role of Polyamines in Autoimmunity: Differences in Cell Types and Disease Models

### 2.1. A Controversy Regarding CD4+ T-Cells

The first indications that polyamines exert a role in autoimmune diseases was described in a pair of papers examining the polyamine inhibitor (2R,5R)-6-heptyne-2,5-diamine or difluoromethyl ornithine (DFMO) on T-cells in lupus-like disease in mice [25,26,27]. In these initial studies, inhibition of polyamine synthesis led to a reduction in T-cell proliferation and lupus-like symptoms in mice. Subsequent literature has hypothesized various mechanisms regarding the role of polyamines in lupus pathogenesis [28,29], but no evidence has emerged to support these theories. Recently, polyamines have been shown to contribute to T-cell-mediated autoimmunity. A study highlighted the central role of polyamine metabolism on the pathogenic fate of Th17 cells in autoimmune pathology [30]. In this work, the authors found that pathogenic Th17 cells have an upregulated polyamine metabolism and that genetic knockout or pharmacologic inhibition of polyamine metabolism resulted in attenuation of autoimmunity. As the Th17/Treg fate is reciprocally regulated [31], the authors also found that inhibition of polyamine synthesis resulted in a skewing towards a regulatory T-cell phenotype, supporting their protective role in autoimmunity. Another recent study supports this phenomenon, showing coordinated blockade of polyamine uptake and synthesis prevented T-cell-mediated inflammation in mouse models of experimental autoimmune encephalomyelitis [32] (EAE). Together, these results indicate that polyamines work to promote the induction of autoimmune-like phenotypes caused by T-cells (Figure 1A, right panel).

Conversely, another impactful publication by the Pearce laboratory indicated that polyamine metabolism plays a key role in regulating the helper T cell (Th) epigenome to prevent autoimmunity [33]. They determined that the prevention of polyamine synthesis through ODC1 inhibition results in disturbed Th lineage fidelity and the induction of an irregular inflammatory T-cell phenotype. Mechanistically, the lack of polyamines prevents EIF5a hypusination, leading to perturbation of the proper acquisition of Th phenotypes. This was further validated by knocking out the gene responsible for hypusination in CD4+ T-cells, resulting in uncontrolled inflammation (Figure 1A, left panel). Another study suggests the anti-inflammatory role of polyamines in T-cells and indicates that spermidine can induce anti-inflammatory responses by enhancing Treg cell development [34]. In this study, they found that spermidine administration induced Treg development from Naive CD4+ lymphocytes, which was regulated by autophagic flux. The mixed results from these studies demonstrate how little is known about polyamine biology in T-cell pathophysiology.

It is clear based on the studies above that there is significant controversy into the roles polyamines play in the context of CD4+ T-cell-dependent immune responses. While precise answers will require significantly more research, there are discrepancies in the experimental approach that can contribute to the differences in results. In Puleston [33] and Wagner [30] et al., they both show that in vitro CD4+ polarization is effected by ODC1 knockout, whereas Wu et al. [32] show that natural levels of Th subsets and Tregs are unchanged in vivo, suggesting that ODC1 may control CD4+ Th differentiation rather than development. Further support comes with the fact that the adoptive transfer of naïve CD4+ ODC1 KO T-cells into Rag^0/0^ mice resulted in colitis. The ablation of the enzyme responsible for polyamine-mediated hypusination (DOHH) resulted in a total loss of Th-lineage fidelity leading to colitis. However, Wagner et al. demonstrated that CD4-perturbation of ODC1 led to a deficiency of Th17 polarization in vitro, and instead, they observed a shift towards the Treg phenotype. Supporting this observation, pharmacologic inhibition of ODC1 led to a decrease in EAE pathology. These results are similar but somewhat at odds with Wu et al. in that they had to coordinately block both polyamine synthesis and uptake to inhibit an EAE phenotype [32]. The fact that CD4 deficiency of polyamine metabolism leads to uncontrolled inflammation in the gut [33], whereas the same deficiency inhibits CNS-inflammatory pathology, which suggests that polyamine metabolism may play divergent roles in Th-function depending on the location of the disease.

### 2.2. CD8^+^ T-Cells

There is significantly less information known about how polyamine metabolism influences CD8^+^ T-cell functionality (Figure 1B). The critical insight came from a recent study where authors performed an in vivo CRISPR screen on antigen-specific CD8^+^ T-cells to examine regulators of T-cell infiltration into triple-negative breast cancer (TNBC). In this study, they found that ODC1 was one of their top guides enriched in tumoral CD8^+^ T-cells, indicating a loss of ODC1-enhanced antigen-specific T-cell killing of tumors [35]. Surprisingly, they found that knockout of ODC1 led to increased degranulation by CD8^+^ T-cells, leading to enhanced tumor clearance. When comparing these results to the papers examining ODC function in CD4^+^ T-cell immunity, it is clear there is a compartmental difference in how polyamines influence different T-cell subsets.

Unsurprisingly, arginine is the principal carbon donor of T-cell polyamines [32,33]. In these works, the authors demonstrated that arginine catabolism supports T-cell activation, proliferation, and differentiation through polyamine biosynthesis. However, these same studies indicate glutamine is a minor carbon donor of polyamine synthesis in T-cells [32,33]. As Myc is one of the top upregulated transcription factors known to elevate glycolysis and glutaminolysis in activation-induced T-cell growth and proliferation, this suggests glutaminolysis may increase polyamine levels to meet T-cell proliferation requirements [36].

### 2.3. Myeloid Cells

While the evidence for the role of polyamines in T-cell function related to inflammation and autoimmunity are varied, the role of polyamines in monocytes/macrophages is better defined (Figure 1C, left panel). The first study to examine this in detail, Zhang et al. demonstrated that spermine addition could potently inhibit LPS induction of TNF-a, IL-1, IL-6, CCL3, and CCL4 in human monocytes/macrophages [37]. Subsequent work by the same scientists found that monocytes/macrophages upregulated polyamine synthesis to prevent TNF-α and nitric oxide (NO) production [38], supporting previous work demonstrating that polyamines inhibit NOS activity in the rat CNS [39]. From a pathophysiological perspective, another group demonstrated the anti-inflammatory role of spermidine on myeloid cells in mouse models of experimental autoimmune encephalomyelitis (EAE), an experimental corollary to multiple sclerosis. In this study, spermidine administration in drinking water was reported to alleviate EAE symptoms by decreasing the expression of Arg1 in macrophages and pro-inflammatory cytokines. [40]. These authors also found that polyamines inhibit NO generation by bone marrow-derived macrophages (BMDMs), supporting a previous study that demonstrated that polyamines could prevent inflammatory activation of microglia by LPS [41]. These studies may explain why macrophage depletion was initially described to prevent EAE induction in mice [42].

### 2.4. Microglia

Another myeloid immune subset involved in autoimmunity, particularly of the CNS, is microglia. Microglia are myeloid cells derived from yolk-sac progenitors that are long-lived in CNS tissues [43]. The specific role of microglial polyamine metabolism in autoimmune pathology is not known; however, in a model of LPS-induced CNS pathology, spermidine was shown to be critical in limiting inflammation [41]. In recent studies of retinal inflammation, authors found that pharmacologic inhibition of polyamine catabolism prevents microglial activation and subsequent retinal vascular damage [44,45] (Figure 1C, right panel). Importantly, while microglia are developmentally distinct from circulating myeloid cells, the experimental ablation of microglia results in the infiltration of peripheral myeloid cells with a similar (but not identical) transcriptomic program as the original microglia [46]. Therefore, the role of polyamines in microglial inflammation may largely overlap with their roles in peripheral myeloid cells.

### 2.5. Dendritic Cells

The role of polyamines in preventing myeloid-driven inflammation has recently been extended to the functionality of dendritic cells (DC) (Figure 1E). Under typical inflammatory conditions, type 1 interferons stimulate dendritic cell activation, which promotes Toll-like receptor 7 (TLR7) upregulation in IFN primed DCs (IFN-DCs) and leads to further inflammation [47,48]. To counter this activation, polyamines prevent glycolytic phenotypes and resultant over-activation of DCs primed by IFN. This paper uniquely found that upregulation of transcription factor FOXO3 was responsible for limiting DC activation, attenuating disease severity in a mouse model of psoriasis [49]. In another recent study, Mondanelli et al. found that Arg-1-mediated the generation of polyamines was necessary for the induction of Indoleamine 2,3-dioxygenase (IDO1) expression in DCs, in which IDO1 is a prototypical effector of immunosuppression [50]. In this study, exogenous polyamines produced by MDSCs were sufficient to induce immunosuppression of DCs in models of skin inflammation. Mechanistically, this was attributed to the activation of SRC kinase by polyamines which promotes IDO1-mediated immunosuppressive activities. In a subsequent review, the authors synthesize the aforementioned studies and demonstrate a positive feedback loop connecting both IDO and polyamine metabolism [51]. They propose that polyamines generated from the TME induce IDO expression, which generates kynurenines that bind to the aryl-hydrocarbon receptor (Ahr). Ahr signaling results in upregulation of both FOXO3a and ODC1, leading to a positive feedback loop of immunosuppression. As most tumors overexpress Myc, which is known to promote both AhR and ODC1 activity, this suggests tumors may be the “spark” by which immunosuppression is ignited in DCs. Conversely, the inability to activate this pathway may underlie autoimmune pathology promotion by DCs. A comprehensive overview of each study and their relevant observations can be found in Appendix A.

### 2.6. Clinical Observations of Polyamines in Autoimmunity

Human clinical data also suggest that polyamine levels are decreased in autoimmune disease and inflammation. Polyamine levels of N1-acetylcadaverine, spermidine, N1-acetylspermidine, and spermine are significantly reduced in Systemic lupus erythematosus (SLE) patients when compared to healthy controls [52]. In addition, comparisons of polyamine metabolites in Graves’ disease (GD), Hashimoto’s thyroiditis (HT), and thyroid autoantibody-positive (pTAb) patients suggest that low spermine is related to thyroid autoimmunity [53]. In summation, decreases in polyamine synthesis or bioavailability can lead to both inflammation and autoimmunity. While myeloid cells’ inflammatory phenotypes appear to be uniformly inhibited by polyamines, their role in T-cell-mediated autoimmunity is still under investigation.

## 3. The Role of Polyamines in Cancer Immunosuppression

### 3.1. Polyamines Are Elevated across Most Cancers

Considering the observation that polyamines act to prevent autoimmune inflammation, one could propose that this phenomenon is occurring in tumors to prevent anti-tumor immunity. Indeed, there is a wealth of literature over the past 40 years to indicate that polyamines are relevant in malignancies. Original studies from the 1980′s demonstrated the increased amounts of polyamines in patients with malignancies when compared to healthy patients. Specifically, researchers found increased polyamine levels in the CSF of brain tumor patients [54], colon cancer tissues [55], and blood/plasma of melanoma patients [56]. However, measurements of polyamine levels in the circulating fluids are unreliable, and later studies have measured ODC1 activity as a proxy for the polyamine metabolic phenotype in brain tumors [57], colorectal cancer [58], breast cancer [59,60], endometrial cancer [61], and neuroblastoma [62]. In fact, many of these studies identify a positive correlation between disease severity and ODC1 levels [58,61,62]. One mechanistic underpinning of ODC1 upregulation in malignant cells is that its activity is directly regulated by Myc [63], a transcription factor overexpressed in many cancers. In fact, Myc overexpression is a central determinant in the oncogenesis of neuroblastoma [62], which may explain the strong connection between Myc and polyamine biology in these tumors. An overview of polyamine measurements in both autoimmunity and cancer in humans can be found in Appendix A.

### 3.2. Pharmacologic Inhibition of Polyamines Is an Effective Chemo and Immunotherapy for Solid Tumors

Considering the significant role of polyamines in tumor biology, pharmacologic and genetic targeting of this axis has been aggressively utilized as a method for treating tumors. The most utilized inhibitor of polyamine biosynthesis is difluoromethylornithine (DFMO), an ornithine analogue originally utilized to treat various trypanosoma species in African sleeping sickness [64]. Many studies have attempted to utilize polyamine blockade as antineoplastic therapy, in which robust anti-tumor results typically occur in combination with polyamine uptake inhibitors [23,65,66]. The explanation for the inclusion of polyamine uptake inhibitors in DFMO administration is because polyamines have a series of transporters dedicated to uptake in cells [67], which can rescue the effects of DFMO administration [68,69]. In neuroblastoma, recent work has shown that the combination treatment of polyamine uptake inhibitor and DFMO results in a dramatic reduction in tumor size in xenograft models [70], leading to the initiation of a number of clinical trials (ClinicalTrials.gov Identifiers: NCT02395666, NCT04301843, NCT02679144) and publications [71,72]. While these studies mainly focus on the direct antineoplastic effects of polyamine blockade, several studies, including our own, have found that polyamine blockade leads to an increase in CD8^+^ T-cell infiltration and inflammatory phenotype in tumors [73,74,75].

In the first seminal study, Hayes et al. revealed that treatment with a combination of polyamine uptake (DFMO) and synthesis (AMXT-1501) leads to a reversal of immune suppression in mouse models of melanoma [74]. In a subsequent study of mouse models in melanoma and colon carcinoma cancer, they showed that treatment with a polyamine blockade resulted in increased Granzyme B^+^ Ifn-γ^+^ CD8^+^ T-cell infiltration and a concomitant decrease in immunosuppressive myeloid cells [75]. These results are consistent with the results of our recent study where polyamine blockade promoted CD8^+^ T-cell infiltration and worked in concert with anti-PD1 or anti-PDL1 immunotherapy to further promote animal survival in mouse models of glioblastoma [73]. The combinatorial effect of polyamine blockade and checkpoint immunotherapy was also demonstrated in mouse models of mammary carcinogenesis [76]. The results of these studies clearly demonstrate that polyamine blockade can alter the TME to promote immune functionality. An overview of the results for polyamine therapy in cancer can be found in Appendix A.

### 3.3. Pharmacologic Blockade of Polyamine Metabolism Targets Immunosuppressive Myeloid Cells in Tumors

A mechanistic explanation for the phenomenon of polyamine blockade inducing anti-tumor immunity has come from recent insights into the immune microenvironment of solid tumors. A hallmark of the tumor microenvironment is the recruitment of immunosuppressive myeloid cells (also called Tumor-associated macrophages—TAMs; Myeloid-derived suppressor cells—MDSCs; or Monocyte-derived macrophages—MDM, depending on the publication), which are a pleiotropic and genetically diverse set of innate immune cells found in most solid tumors [77]. These cells exist across a functional spectrum, with one end being pro-inflammatory and the other end being reparative or immunosuppressive. This topic has been extensively reviewed in previous publications [78,79,80]. It comes as no surprise that within tumors, myeloid cells exist typically towards the immunosuppressive edge of this spectrum [77], particularly in CNS tumors, where they are the most abundant infiltrating immune cells [81,82,83,84]. Canonically, the reparative/immunosuppressive phenotype is defined by cellular expression of Arginase-1 [85] (Arg1), which generates ornithine from arginine [86]. The enzymatic reactions catalyzed by Arg1 and Glycine Amidinotransferase [87] (Gatm) are the sole sources of ornithine generation in cells, which is then catalyzed by ODC1 to produce the initial polyamine putrescine [88]. Therefore, the identity of the reparative/immunosuppressive phenotype of myeloid cells is directly linked to polyamine generation (Figure 2A).

To determine if polyamine generation underlies the immunosuppressive or “reparative” phenotype of myeloid cells, the Wilson laboratory generated conditional knockouts of ODC in myeloid cells utilizing LysM-Cre as a driver of ODC-flox-mediated recombination [89]. These mice exhibit more pro-inflammatory macrophages in the gut [89], also shown in models of colitis-associated carcinogenesis [90]. Similarly, we have shown a specific reduction in TAMCs in murine models of GBM treated with polyamine inhibition [73]. Together, these data point to the effects of polyamine inhibition on the myeloid component of the TME.

What exactly polyamines are doing in the myeloid cells to promote immune suppression is not concretely established, but there are several viable theories to explain why they are so important for myeloid function (Figure 2). One prominent hypothesis is the metabolic competition for arginine between immune cells in the tumor (Figure 2B). Geiger et al. demonstrated that arginine uptake is critical for T-cell proliferation and effector functions [91] and that arginine depletion induces further MDSC recruitment while also inhibiting T-cell proliferation [92]. A second possibility is the transfer of polyamines from myeloid cells directly to tumors (Figure 2C). As polyamines are fundamentally required for cellular proliferation, myeloid cells may simply act as feeder cells for growing tumors. There is substantial evidence that polyamines are actively taken up by malignant tissues, which is particularly relevant during the inhibition of de novo polyamine synthesis [66,70,93]. A third possibility comes from the Pearce laboratory, where they found that polyamine-mediated hypusination of EIF5a directly controls mitochondrial transcription and translation of proteins which control the electron transport chain (ETC) and tricarboxylic acid (TCA) cycle [94] (Figure 2D). In this study, they argue that polyamines made by IL-4 induced macrophages maintain the metabolic phenotype of an alternatively activated macrophage by promoting oxidative phosphorylation and the TCA cycle. Finally, the basic nature of polyamines allows myeloid cells to buffer themselves in the acidic pH environments present in the TME, promoting metabolic and immunosuppressive phenotypes (Figure 2E). Our lab has shown this phenomenon in glioblastoma [73], consistent with previous work that has shown polyamine pH buffering in leukemic cancer cells [95]. It is likely that all these processes occur simultaneously in varying degrees within the tumor environment.

## 4. Do Polyamines Prevent Autoimmune-Mediated Eradication of Tumor Tissues?

As discussed, polyamines play an important immunoregulatory role in the homeostasis of tissues. While not always consistent, most research indicates the presence of increased polyamines to be associated with the prevention of autoimmunity, inflammation, and brain-associated pathologies. Similarly, the presence of increased polyamines is associated with immunosuppression and tumor progression. This begs the question: if polyamine blockade can promote anti-tumor immune responses, is it doing so via the promotion of autoimmunity? Indeed, the question of whether autoimmunity is required for anti-tumor immunity is still an ongoing issue of debate. An early study performed by Miska et al. [96] examined this question utilizing two different models of autoimmunity/cancer in which T-cells had specificity for both normal tissue (pancreas) and tumor (pancreatic or melanoma cell lines). In these experiments, they found that diabetogenic T-cells were efficient at destroying islets while tumors were able to use regulatory T-cells (Tregs) to prevent their own destruction. Importantly, treatment with Anti-CTLA4 overcame this tumor resistance and promoted tumor regression. The results of this work indicate that some level of autoimmunity may be required to elicit potent anti-tumor immunity.

In seminal NEJM studies on combinatorial checkpoint immunotherapy for melanoma (CTLA4 ^+^ PD1 blockade), it was found that immune adverse related events (iAREs) were significantly enhanced in dual therapy patients, resulting in discontinuation of combinatorial therapy [97,98]. The types of immune adverse events were diverse, with grade 3 or grade 4 colitis, diarrhea, and liver ALT levels being the most prominent. After these trials, multiple studies have shown that patients with the most iAREs have the best prognostic outcomes. The positive connection between iAREs and progression-free survival (PFS), Overall Response Rate (ORR), and overall survival (OS) is supported by multiple studies. For example, work performed by Berner et al. showed PD-1-elicited dermatological iAREs were significantly more frequent in patients with complete or partial remission (68.2%) compared to those with progressive or stable disease (19.6%) [99]. This trend has been reproducible and is mostly associated with blockade of the PD/PD-L1 axis rather than CTLA4 blockade [100,101]. For further reading, an in-depth review of checkpoint blockade and iAREs has been recently published [102].

These iAREs can either be a result of a directed T-cell response against shared tumor/normal tissue antigens or a broad and non-targeted increase in inflammation. Dissecting the two possibilities is a challenging effort, but there is evidence for both processes occurring. One severe iARE, myocarditis, indicates a role for both T-cells and inflammatory monocytes [103]. The work by Berner et al. supports the first hypothesis, as they found nine antigens shared between both the tumor and skin tissue [99]. Furthermore, many iAREs are associated with B and T-cell responses [104,105]. However, some studies also point the finger at myeloid-driven inflammation as a source of iAREs. A recent study indicated that PD-L1 blockade, contrary to PD-1 blockade, induced an inflammatory phenotype of myeloid cells [106]. This idea is supported by a study of responder vs. non-responders in metastatic melanoma, in which the authors show that myeloid-attracting chemokines are upregulated in non-responding patients [107]. Overall, these data support the notion that the myeloid monocyte compartment is critical to the effectiveness of immunotherapy for tumors. Considering the influence of the myeloid compartment on immunotherapeutic efficacy, it is possible the levels of circulating (or tumoral) polyamines might act as metabolic determinants of responsiveness to immunotherapy. Furthermore, considering there are several clinical trials utilizing polyamine inhibition as a cancer therapy, emerging data suggest these therapies may benefit from examining their effects on the immune landscape, as it may provide critical clues as to the utility of immunotherapies in cancer.

In summation, these studies demonstrate a cycle in which polyamines control the balance of inflammation and immunosuppression (Figure 3). When an inflammatory stimulus is generated, myeloid cells are recruited from the blood and exhibit a pro-inflammatory phenotype characterized by lymphocyte activation and tissue destruction (Figure 3A). The result of ongoing inflammation and apoptotic-body clearance results in the immune suppressive (or pro-resolving) phenotype of myeloid cells (Figure 3B). This phenotype is characterized by immunosuppressive signals and polyamine generation [108]. Thus, in autoimmunity, if polyamine generation is insufficient, myeloid cells may be unable to restrain inflammation. Conversely, many tumors promote the polyamine metabolic phenotype, and the continuous recruitment of myeloid cells by the tumors [109,110] facilitates both tumor growth and therapeutic resistance [111,112,113,114] (Reviewed extensively in [115,116,117]) (Figure 3C). Thus, a polyamine blockade may a relevant strategy to enhance immunotherapeutic efficacy, possibly through the enhancement of autoimmunity.

## 5. Limitations and Future Directions

A significant limitation of these studies is the discordance between the observation of exogenous/circulating polyamines and intracellular polyamine biology. Clinical observations of polyamine levels in autoimmunity are mainly in systemic circulation without direct measurement of tissues. Therefore, there is a distinct lack of understanding regarding the contribution of exogenous polyamines compared to intracellular polyamines in both auto-immune pathologies. Future work will need to be conducted to examine how different subsets of immune cells respond to polyamine blockade in vivo and how this relates to disease outcomes.

Another fascinating limitation of previous work is that there appear to be other pathways of polyamine synthesis in mammalian tissues that cannot be attributed to typical biosynthetic pathways associated with agmatine metabolism [118]. In the future, uncovering these mechanisms and understanding how it is involved in immunity will be critical to our understanding of how polyamines control immunity. Related to this limitation is a lack of consensus regarding polyamine transport in cells. There have been several potential transporters indicated in the transport of polyamines, and they differ based on which polyamine is measured, whether import or export is being tested, and what cellular population is being studied [67,119,120]. A complete understanding of polyamine transporters and the context in which they function (particularly in malignancy) will be critical in future work aimed at modifying the polyamine metabolism as a therapeutic approach.

Another limitation of previous research is the degradation of exogenous polyamines by degradative enzymes. Recent studies have demonstrated that the copper-containing bovine serum amine oxidase in fetal bovine serum could oxidize exogenously added polyamines and results in cytotoxicity-inducing products, including hydrogen peroxide, ammonia, and reactive aldehydes [121]. Importantly, this is known to induce cellular toxicity [122], which may explain the previously reported role of polyamine in inducing cytotoxicity in vitro [123,124]. In the future, scientists must consider these effects when interpreting in vitro exogenous polyamine Appendix A.

## 6. Concluding Remarks

It is fascinating that despite being the first metabolites ever discovered, there is still so much information to learn about polyamines. Even less is known about how polyamine levels may act as a proxy for understanding the immunological state of a disease or therapeutic response. Despite our limited knowledge, it is clear that polyamine metabolism plays an essential role in the balance between autoimmunity/inflammation and immunosuppression. This work suggests that monitoring polyamine levels either in circulation or in tissues may be used as a prognostic marker for both autoimmunity and immunotherapy. Perhaps increased levels of circulating polyamine may portend resistance to immunotherapies. Or conversely, reduced polyamine levels may indicate a patient will have more severe iAREs when being treated for immunotherapies. In summation, the measurement of polyamine metabolism may be a valuable tool to understand a broad spectrum of human immunology and responses to therapies for disease.

## Figures and Tables

**Figure 1 cells-11-00896-f001:**
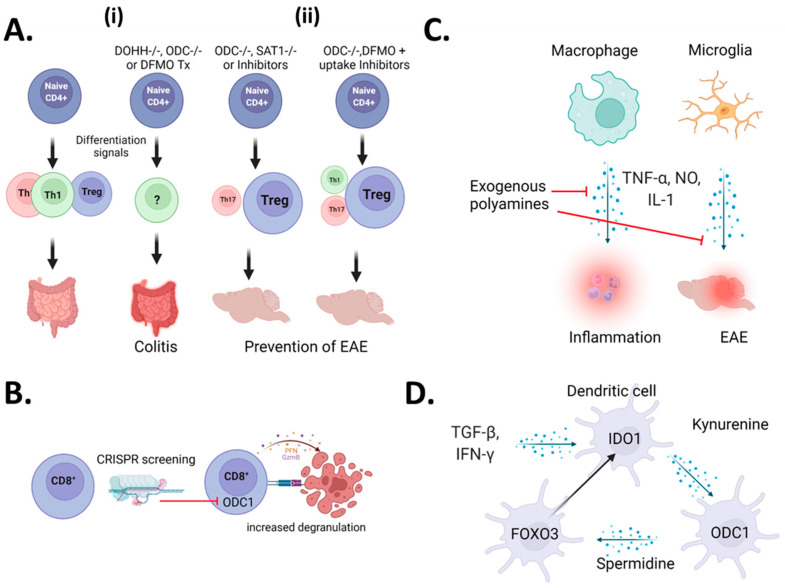
Overall role of polyamines in inflammatory processes of immune subsets. Sparked by three high-impact publications, the role of polyamine on CD4+ T-cell physiology is highly controversial (**A**). In the first publication, (i) blockade of polyamine synthesis led to unstable lineage fidelity, resulting in uncontrolled inflammation. In the other publications, (ii) polyamine blockade led to the amelioration of experimental autoimmune encephalomyelitis (EAE) by preventing inflammatory Th17 pathogenesis and inducing a regulatory T-cell phenotype. In the only study to date that directly examines polyamine function in CD8+ T-cells, the authors found thorough CRISPR screening that the deletion of ODC1 led to enhanced cytotoxicity and the degranulation of T-cells (**B**). In myeloid lineage, cells and microglia polyamines are uniformly associated with inhibition of inflammation and autoimmune phenotypes (**C**). In dendritic cells (DCs), a metabolic positive feedforward loop instigated by IFN-y and/or TGF-b results in a tolerogenic phenotype (**D**). Blockade of many aspects of this pathway in DCs leads to increased autoimmune inflammation. Figure created with BioRender.com.

**Figure 2 cells-11-00896-f002:**
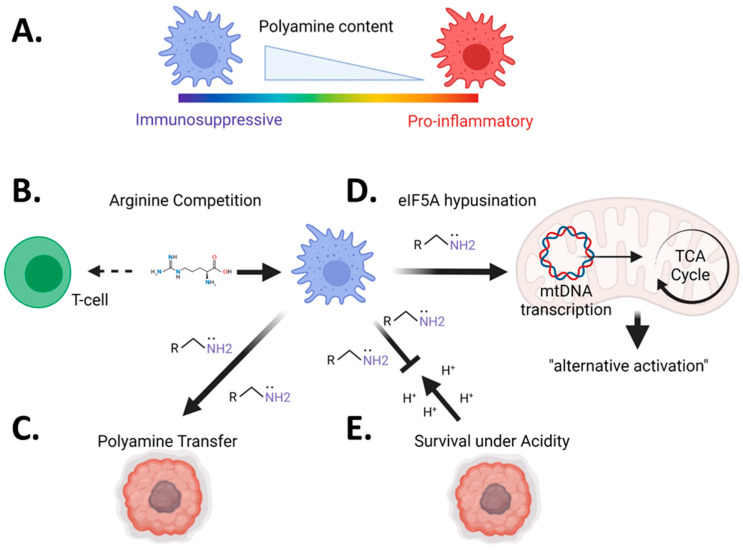
Myeloid-linked mechanisms by which polyamines promote immunosuppression in cancer. While the mechanisms are still being uncovered, there are several potential mechanisms for why polyamine accumulation in myeloid cells is immunosuppressive (**A**). One hypothesis is that the propensity of alternatively activated macrophages to consume arginine, the precursor to polyamines, deplete arginine from the inflammatory microenvironment sequestering it from T-cells that need it for activation/function (**B**). Another proposed mechanism is that polyamine generation by myeloid cells directly feeds tumors as they actively take-up polyamines (**C**). Another mechanistic study indicated that polyamines drive mitochondrial gene expression and promote the metabolic phenotype associated with alternatively activated macrophages (**D**). We have recently shown that polyamine can be used as a mechanism to buffer intracellular pH to allow for the survival and function of myeloid cells in brain tumors (**E**). Despite the growing list of mechanisms by which polyamines promote immune suppression in myeloid cells, much of the biology is still yet to be determined. Figure created with BioRender.com.

**Figure 3 cells-11-00896-f003:**
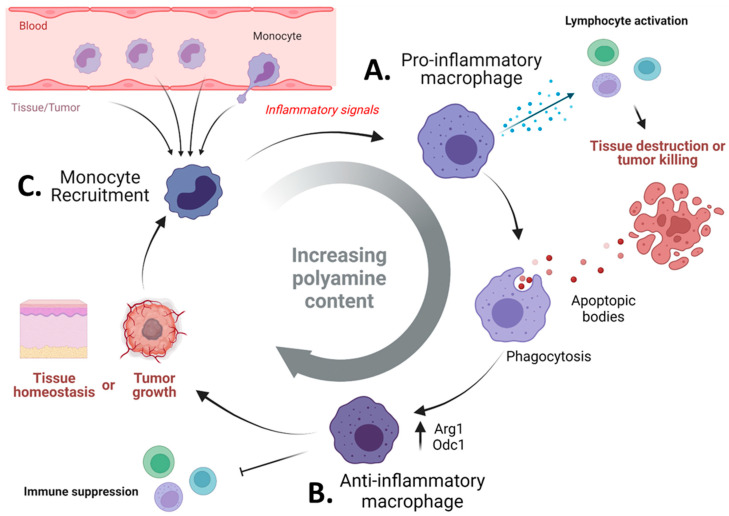
Polyamines maintain the delicate balance of immunity and tolerance by myeloid cells. In typical scenarios, inflammation is a transient process, by which immunogenic signals eliciting a response are cleared. In order to slow down these processes, the immune system has several internal checkpoints meant to prevent uncontrolled immunity from occurring. These processes occur over time, with an initial wave of inflammation followed by upregulation of checkpoints designed to put the brakes on the immune system (**A**). For myeloid cells, this switch to a pro-resolution phenotype is associated with arginase-1 metabolism, ODC1 activity, and ultimately polyamine synthesis. This response is stimulated by the phagocytosis of apoptotic bodies, which act as a source for arginine and nucleotides for these cells to suppress and proliferate, respectively (**B**). It is likely that tumors take advantage of this process as the tumor continues to grow by providing chronic inflammatory signals to recruit myeloid cells and then providing apoptotic bodies (and other immunomodulatory signals) to promote their M2-like or pro-resolving phenotype (**C**). From the perspective of polyamines, they increase as the cell goes from an inflammatory phenotype to a pro-resolution or immunosuppressive phenotype. Thus, a deficiency in polyamine levels may perpetuate inflammation, while conversely, excessive levels promote tumorigenesis and immunosuppression. Figure created with BioRender.com.

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
