# Peer review of "Polyamine Immunometabolism: Central Regulators of Inflammation, Cancer and Autoimmunity"

_cells, 2022, doi:10.3390/cells11050896_

Round 1

Reviewer 1 Report

TO THE AUTHORS:

Major critics:

The review of Tzu-Yi Chia, Andrew Jacob Zolp and Jason Miska Submitted to section: Cellular Immunology, (Immunometabolism: A Therapeutic Target in Cancer and Inflammatory) of the journal “Cells” ID: cells-1598926, entitled  “Polyamine immunometabolism: central regulators of inflammation, cancer and autoimmunity”, is one of the extremely important and interesting for cancer and polyamine research and for immunology as well as for neuroscience since for example, glioblastoma is one of the most aggressive and fatal brain cancers due to the highly invasive nature of glioma cells. However, very important role of the tri-partite cell ensemble such as microglia-astrocytes-glioma is missing as well as other important points. After revision, the manuscript may be highly recommended for publication and will be promptly cited in future.

Authors specifically highlighted that “…the efficacy of immunotherapies are linked to autoimmune consequences in humans, termed immune-adverse related events (iARE). Therefore, polyamine levels may govern the inflammatory response to immunotherapy”. Authors built the review focusing on the combinatorial treatment with polyamine intervention in combination with an immunotherapy. However, one type of the cells participating in immunoresponses, the macrophage lineage, so called microglia (that are residing in the brain) may even help glioblastoma to invade brain parenchyma (Rolón-Reyes et al., 2015).

Therefore, authors need to improve this Review adding a novel conceptual extend and a proper literature about fundamental findings that polyamines (i) are accumulated in brain in astrocytes but not in neurons (Laube and Veh, 1997) and in (ii) other glial cells of ectodermal origin (Biedermann et al., 1998; rev. Skatchkov et al., 2014; 2016), while (iii) microglia as macrophages are from mesoderm, and polyamines may simply kill microglia (Takano et al., 2003) and neurons (Sparapani et al., 2008) if (iv) there are no astrocytes being placed around (.  This is important for a novel envision of current search on the efficacy of anti-cancer therapy in pre-clinical models of cancer related to polyamine turnover. 

There is a widespread understanding that polyamines are neurotoxic and play no physiological role in the brain. However, when synthesis of spermine is downregulated due to a mutation in the X-chromosome, it causes Snyder-Robinson Syndrome that severely affects human brain function leading to mental retardation, hypotonia and cerebellar circuitry dysfunction (Cason et al., 2003; Ikeguchi et al., 2006) and many studies have reported both neuroprotective effects of polyamines (Gilad and Gilad, 1991; Ivanova et al., 1997; Zhang et al., 1997; 1999; Ferchmin, et al., 2000; Tracey, 2002; Bell et a., 2007; Noro et al., 2015; Sigrist et al., 2018), in addition to clearly neurotoxic effects (de Vera et al., 2007).

Opposing consequences of polyamine neuroprotection may be when aggressive radicals are release in the brain that actually depends on a modification of the base polyamines to radicals.  Ischemia can cede a cascade of cytotoxic molecules responsible for the death of viable cells in the brain. Under these conditions, polyamines may be oxidized and converted to cytotoxic aldehydes and reactive oxygen species, these may lead to neurotoxicity (damaging proteins, DNA and lipids). Conversely, unmodified polyamines and their analogs may block Ca2+-permeable receptors and channels (NMDAR, AMPAR-GluR2-lacking), underlying a potential protection of neurons from apoptosis via calcium overload (Abushek et al., 2013).

Despite these recognized modifications and potential targets, the localization and dynamics of polyamines in the brain are largely understudied. As discussed in detail (Skatchkov et al., 2016), new findings indicate that polyamines are normally highly localized in the glia, not in neurons. However, macroglia called neuroglia such as astrocytes, Muller and Bergmann glia, do not synthesize spermidine (SD), a precursor of spermine (SP), in normal condition (Krauss et al., 2006; 2007), and instead accumulate polyamines from the extracellular medium (buffering), and release them under conditions of depolarization or metabolic downregulation. Many authors therefore described anti-inflammatory and antioxidant action of polyamine (an example: Belle et al.,2004; Soda, 2022) and as increasing life span metabolites (Eisenberg et al., 2009; Viltard et al., 2019).

Since even after blockade of ornithine decarboxylase (ODC) by di-fluoromethyl-ornithine (DFMO), the synthesis of polyamine putrescine (PUT) and consequently production of PUT, spermidine (SPD), spermine (SPM) was not stopped, this tells that (i) either other enzymes can overcome the PA-synthesis limit by ODC (such as agmatinase) and in fact help to produce putrescine by another way (Laube and Bernstein, 2017; Polis et al., 2022) or (ii) the cells have SLC subfamilies of polyamine transporters to gain PUT and SPD, SPM supply (Sala-Rabanal et al., 2013; Malpica-Nieves et al., 2020; 2021). The arginine-derived polyamine synthesis is the alternative way to produce polyamines by glioblastoma if not ODC is involved. These papers should be cited (see below).

Arginases and agmatinases are enzymes involved in polyamine machinery (Swensson et al., 2008; Bernstein et al., 2011; Peters et al., 2013; Laube and Bernstein, 2017; Polis et al., 2017), therefore the interest to PA research is highlighted, specifically in aging (Minois et al., 2011; 2014; Madeo et al., 2018; 2020; Viltard et al., 2019) and diseases. The authors of current research, being specialists in cancer research and immune-system specifically need to combine actually all “pro” and “contra”  arguments about polyamines to describe glioma and glioblastoma interactions with polyamines and astrocytes-microglia influence. This is because the manuscript is the Review that should bring the interest of readers and trigger good citing of the presented set of crucial data for the benefit of the journal and authors. 

Interestingly, it was shown that polyamine spermidine is anti-cancer agent and is providing extended life span and improve cognitive and memory function, thus improving longevity via autophagy pathway (Eisenberg et al., 2009; 2016; Sigrist et al., 2014; Schroeder et al., 2021). Therefore not just PAs can be harmful but beneficial. Intriguingly, Viltard et al. (2019) found that acetylated forms of PAs are accumulated in naked mole-rat (NMR). This highlights that NMRs, (extremely long living animals; about 20 time longer than lab rats), differ from lab rats and cancer animal models in the degradative PA machinery.

Specifically, in the relation to glioblastoma, which currently is not having promising treatment, the authors need to discuss some previous findings as well as novel. For example, the authors introduce readers to polyamine support of proliferation (lines 47-52) without citing the recent finding of Malpica-Nieves et al., (2020) in glial cells showing that after a block of polyamine synthesis in healthy juvenal astrocytes the supplement of spermidine rescue astrocytic survival and proliferation.

This is very important that polyamines not always are proinflammatory but rescuing factors. However, if the uptake of polyamines is blocked together with the synthesis then the proliferation stopped completely in glia (Malpica-Nieves et al., 2020) and that is probably why glioblastoma internalizes organic cation transporter, SLC22A type, as was shown by Kucheryavykh et al., (2014). Since authors have concerned potassium inwardly rectifying channels (Kir channels) which are highly sensitive to polyamines, it should be stated in the review that Kir4.1 glial channels are also internalized in glioma (Olsen and Sontheimer, 2008). Such loss of functional Kir channels triggers re-entry of cells into the cell cycle and reactive gliosis and has been shown in a number of neurological diseases including malignant gliomas. Ion addition, such Kir4.1 loss lead to temporal lobe epilepsy, amyotrophic lateral sclerosis, retinal degeneration. Conversely, expression of Kir4.1 is sufficient to arrest the aberrant growth of these glial derived tumor cells. Therefore, both SLC22A3 and Kir4.1 represent a potential therapeutic target in a wide variety of diseases including glioma, astrocytoma, glioblastoma. conditions. These references should be cited.

This transporter is the major carrier of spermidine (Sala-Rabanal et al., 2013) expressed in glia and therefore glioblastoma will not take polyamines and the use of modified polyamines such as cisplatin or semapimod have resulted in a failure of such chemotherapy. Also, that is probably why glioblastoma is resistant to polyamine transport blockers. Therefore such arguments authors need to add (lines 212-215 or lines 259-262, where authors concerned tumor environment such as astrocytes and microglia and polyamine exchange there. Authors also need to review literature to highlight why glioma cells loss gap junctions that widely occur in healthy glial cells where spermine dramatically help propagation of macromolecules in astrocytic syncitium (Benedikt et al., 2012) and protecting gap junction closure (Kucheryavykh et al., 2017). The present Review could be outstanding in many respects.

Adult astrocytes do not synthesize polyamines and Peters et al., (2013) found that arginase (Arg-1) activity is  not present in glial cells which outnumber neurons about 13 time in brainstem and 3.5 times in cortex in human brain (Lent et al., 2012).  So, the glial cells (astrocytes, Muller and Bergmann glia) robustly accumulate polyamines without a synthesis by alternative transport pathway (Malpica-Nieves et al., 2020) and release polyamines via Cx43 hemichannels (Malpica-Nieves et al., 2021).  The role of polyamine interactions in the tri-partite ensemble: microglia (macrophages), astrocytes and glioma should be discussed in this Review.  Proper references need to be included in.

This manuscript can be improved by the notes above and beneath.

In the Introduction or/and in the discussion a role of cellular origin (neurons versus glial cells for example)  needs to be pointed because astrocytes but not neurons accumulate polyamines (Laube and Veh, 1997; Biedermann et al., 1998). This will be for the benefit of the manuscript itself, for the readers, for citations and for the journal.

 It is recommended to include original references:

 Polyamines promote longevity:

Eisenberg,  T., Knauer, H.,  Schauer, A., Büttner, S., Ruckenstuhl, C.,  Carmona-Gutierrez, D., Ring, J.,  Schroeder, S.,  Magnes, C.,  Antonacci, L., Fussi, H.,  Deszcz, L., Hartl, R.,  Schraml, E.,  Criollo, A.,  Megalou, E.,  Weiskopf, D.,  Laun, P., Heeren, G., Breitenbach, M., Grubeck-Loebenstein, B.,  Herker, E.,  Fahrenkrog, B.,  Fröhlich, K.-U., Sinner, F., Tavernarakis,  N., Minois, N., Kroemer, G.  and  Madeo, F. (2009) Induction of autophagy by spermidine promotes longevity. Nat Cell Biol. 11(11):1305-1314);

Wirth A, Wolf B, Huang CK, Glage S, Hofer SJ, Bankstahl M, Bär C, Thum T, Kahl KG, Sigrist SJ, Madeo F, Bankstahl JP, Ponimaskin E. (2021) Novel aspects of age-protection by spermidine supplementation are associated with preserved telomere length. Geroscience. 43(2):673-690. doi: 10.1007/s11357-020-00310-0

Soda K. (2022) Overview of Polyamines as Nutrients for Human Healthy Long Life and Effect of Increased Polyamine Intake on DNA Methylation. Cells. 11(1):164. doi: 10.3390/cells11010164.

Polyamines provide enhanced propagation of molecules in the glial syncitium  

Benedikt J, Inyushin M, Kucheryavykh YV, Rivera Y, Kucheryavykh LY, Nichols CG, Eaton MJ, Skatchkov SN (2012) Intracellular polyamines enhance astrocytic coupling. Neuroreport. 23(17):1021-1025).

Alternative polyamine pathways

Laube G, Bernstein HG. (2017) Agmatine: multifunctional arginine metabolite and magic bullet in clinical neuroscience? Biochem J. 474(15):2619-2640. doi: 10.1042/BCJ20170007.

Polis B, Karasik D, Samson AO. (2021) Alzheimer's disease as a chronic maladaptive polyamine stress response. Aging (Albany NY). 13(7):10770-10795. doi: 10.18632/aging.202928.

Polyamine glial transporter family should be discussed and cited, such as SLC22A1-,2,3 and SLCB1 transporters (Hiasa et al., 2014; Sala-Rabanal et al., 2013). These transporters are polyspecific and may take up not only polyamines but their precursors including agmatine, putrescine, valine, etc. This is specifically important in terms of glioma developments because the arginase activity may be regulated by the level of the final products. 

The accumulation of polyamines in astrocytes, Bergmann and Müller glia has been demonstrated (Laube and Veh, 1997; Biedermann et al., 1998) while the absence of a synthesis of polyamines in glial cells is obvious (Peters et al., 2013; Krauss et al., 2006; 2007; Bernstein and Muller, 1999).  Therefore, authors can cite:

 Sala-Rabanal M, Li DC, Dake GR, Kurata HT, Inyushin M, Skatchkov SN, Nichols CG (2013) Polyamine Transport by the Polyspecific Organic Cation Transporters OCT1, OCT2 and OCT3. Mol Pharm. 10: 1450-8.

Malpica-Nieves CJ, Rivera-Aponte DE, Tejeda-Bayron FA, Mayor AM, Phanstiel O, Veh RW, Eaton MJ, Skatchkov SN. The involvement of polyamine uptake and synthesis pathways in the proliferation of neonatal astrocytes. (2020) Amino Acids. 52(8):1169-1180. doi: 10.1007/s00726-020-02881-w.

Malpica-Nieves, C.J.; Rivera, Y.; Rivera-Aponte, D.E.; Phanstiel, O.; Veh, R.W.; Eaton, M.J.; Skatchkov, S.N. Uptake of Biotinylated Spermine in Astrocytes: Effect of Cx43 siRNA, HIV-Tat Protein and Polyamine Transport Inhibitor on Polyamine Uptake. Biomolecules 2021, 11, 1187. https://doi.org/10.3390/biom11081187

Glioma internalizes Polyamine sensitive Kir channels and transporters, should be cited

Olsen ML, Sontheimer H. (2008) Functional implications for Kir4.1 channels in glial biology: from K+ buffering to cell differentiation J Neurochem. 107(3):589-601. doi: 10.1111/j.1471-4159.2008.05615.x.

Kucheryavykh LY, Rolón-Reyes K, Kucheryavykh YV, Skatchkov S, Eaton MJ, Sanabria P, Wessinger WD, Inyushin M. (2014)  Glioblastoma development in mouse brain: general reduction of OCTs and mislocalization of OCT3 transporter and subsequent uptake of ASP(+) substrate to the nuclei. J Neurosci Neuroeng. 3(1):3-9. doi: 10.1166/jnsne.2014.1091.

Interplay between glioma and microglia and astrocytes

Rolón-Reyes K, Kucheryavykh YV, Cubano LA, Inyushin M, Skatchkov SN, Eaton MJ, Harrison JK, Kucheryavykh LY. (2015) Microglia Activate Migration of Glioma Cells through a Pyk2 Intracellular Pathway. PLoS One. 10(6):e0131059. doi: 10.1371/journal.pone.0131059.

Overall, the manuscript is of excellent quality and timely. It could be improved by adding the recommended paragra[hs into discussion/introduction and citations about the relevance to polyamine uptake versus synthesis in glial cells versus neurons and the role of glial polyamines, polyamine sensitive glial channels and transporters to understand cancer disease.

Overall, the manuscript is of excellent quality and timely. It could be improved by adding the recommended discussion/introduction and citations about the relevance to polyamine uptake for brain function and in diseases in glial cells, microglia, astrocytes, as well as the  absence of synthesis of polyamines in adult glial cells; as well as glial polyamine function versus neuronal and the role of glia and polyamines in glioblastoma disease.

Reviewer 2 Report

Chia et al. highlighted the current studies demonstrating the roles of polyamines in the regulation of the immune system, particularly inflammation (autoimmunity) and cancer immunosuppression. Overall, this topic is well chosen and surely of interest to investigate in the field. However, this article surely needs to be improved in contents and structure, and I have several concerns, as follows. I hope these comments help improve the quality of this article.

  1. This review article focuses on the regulation of the immune system, especially inflammation (autoimmunity) and cancer immunosuppression, therefore, it will be better to remove the 3rd paragraph of the Introduction section (role of polyamines in reproduction). Also, the Introduction section surely needs to include a brief introduction to the immune system focusing on (autoimmunity) and cancer immunosuppression.
  2. Fig. 1 depicting the manifold roles of polyamines in biology is too simple, superficial, and not directly associated with the topics of this article. Therefore, it is better to remove it.
  3. Section 2 (The role of polyamines in autoimmunity) and 3 (The role of Polyamines in Cancer Immunosuppression) are long and not well organized. These sections need to be subcategorized based on disease types or polyamine types etc...
  4. For readers, it will be much better to summarize the study observation (results) of sections 2, 3, and 4 in one table or separate tables (one summary table for each section).
  5. Like section 3, section 2 (The role of polyamines in autoimmunity) also needs to be summarized in a figure.
  6. The authors summarized the studies discussed in this article and made a simple conclusion in the Concluding Remarks section. The reviewer highly recommends describing the limitation of the current studies and the directions of the future studies that need to be further investigated to solve these limitations.
  7. There are many typos and grammatical errors. Please go over the entire manuscript carefully and correct all the typos and grammatical errors.

Reviewer 3 Report

This is an excellent review of the current state of knowledge regarding polyamine metabolism in various physic-pathological settings. The co-authors did a credible job of illustrating the overall concept and detailed key information presented by the literature. As a timely update on a topical field in immunology and cancer research, this review will undoubtedly be a good reference for immunologists and oncologists, and scientists who are just beginning to consider this topic. I understand this review is intended to be a concise update on the field and by no means to cover every aspect. However, I want to encourage the authors to consider the following suggestions to further improve contextual evidence's clarity.

1) There is a significant discrepancy regarding the outcome of genetic/pharmacological ablation of ODC alone in T cells (references 31, 33, and 34). I want to encourage authors to include a brief discussion/highlight on the topic.

2) I want to encourage authors to include a brief discussion on the carbon source of polyamine de novo biosynthesis and the role of Myc in regulating polyamine biosynthesis in immune cells (evidence has been provided in references 31 and 33, and PMID: 22195744 )

3) Within the context of discussing the role of polyamine in controlling autophagy, an essential technical issue (PMID: 32430398, which is commonly missed in the field) needs to be addressed.

4) ODC was identified as one of the top hits in an in vivo CRISPR screening for regulating CD8+ T cell-mediated anti-tumor immune response (PMID: 31442407). 

Reviewer 4 Report

The review by Chia et al. summarizes the current knowledge, and the unavoidable gaps of knowledge, about the relevance of polyamines in the regulation of immune responses and thus their pathogenetic contribution to the development/progression of tumoral and autoimmune diseases. The review is exhaustive and well written.

I woud suggest introducing a reference to the papers of Mondanelli et al. (Immunity 2017 and Trends Immunol 2020), where the contribution of polyamines to the development/maintenance of a suppressive phenotype in DCs, and the intersection between polyamines and kynurenines pathways in the context of neoplasia and autoimmunity are dissected.

Round 2

Reviewer 1 Report

I am satisfied by the responses of the authors of the review (Chia, et al.,“Cells” ID: cells-1598926) to all four reviewers including my critics. The authors should carefully check the list of the references to avoid misspelling and missed letters (see REFs # 14, # 98).

I agree authors cannot cover all requested issues, but the review is substantially improved including new concepts, literature and rewriting several key paragraphs.

Even more, promises the authors highlighted are to write a new review(s) on the points about Central Nervous System (CNS) immune-responses and polyamines in respect to brain cancers, such as astrocytoma, oligodendroglioma, schwannoma, glioblastoma, ependymoma, meningioma, medulloblastoma, lymphoma, neuroblastoma, and other cancers.

 This and future reviews are extremely important in the scope of cancer-immunoresponses-polyamines. It is still little known about the very important role of the tri-partite cell ensemble such as microglia-astrocytes-glioma. The manuscript is now highly recommended for publication and I believe it will be beneficially cited in the future.

Reviewer 2 Report

The authors have appropriately addressed all reviewer's comments, and the article is now acceptable for publication.